# Nanostructured Strontium-Doped Calcium Phosphate Cements: A Multifactorial Design

**Massimiliano Dapporto \*, Davide Gardini** **, Anna Tampieri and Simone Sprio \***

Institute of Science and Technology for Ceramics, National Research Council of Italy, 48018 Faenza, Italy; davide.gardini@istec.cnr.it (D.G.); anna.tampieri@istec.cnr.it (A.T.)
\* Correspondence: massimiliano.dapporto@istec.cnr.it (M.D.); simone.sprio@istec.cnr.it (S.S.)

**Abstract:** Calcium phosphate cements (CPCs) have been extensively studied in last decades as nanostructured biomaterials for the regeneration of bone defects, both for dental and orthopedic applications. However, the precise control of their handling properties (setting time, viscosity, and injectability) still represents a remarkable challenge because a complicated adjustment of multiple correlated processing parameters is requested, including powder particle size and the chemical composition of solid and liquid components. This study proposes, for the first time, a multifactorial investigation about the effects of powder and liquid variation on the final performance of Sr-doped apatitic CPCs, based on the Design of Experiment approach. In addition, the effects of two mixing techniques, hand spatula (low-energy) and planetary shear mixing (high-energy), on viscosity and extrusion force were compared. This work aims to shed light on the various steps involved in the processing of CPCs, thus enabling a more precise and tailored design of the device, based on the clinical need.

**Keywords:** calcium phosphate bone cements; strontium; hydroxyapatite; planetary ball milling; design of experiments; extrusion

## 1. Introduction

In the last decades, the use of self-hardening injectable pastes for bone tissue engineering, such as calcium phosphate cements (CPCs), has been widely considered as a promising approach, due to their ability to mold to the shape of the bone cavity and set in situ upon injection [1]. CPCs are bioactive, especially due to their chemical resemblance with the inorganic component of bone, as well as their nanostructured morphology. In this respect, their intrinsic ability to interact with the surrounding tissue, avoiding both pH and temperature fluctuations during hardening, allows the drawbacks typically reported for bio-inert acrylic cements to be overcome [2–4].

Among the various formulations proposed for CPCs [5], the hydrolysis of α-tricalcium phosphate (α-TCP) phases (i.e., α-Ca$_3$(PO$_4$)$_2$) as a unique cement inorganic precursor is particularly interesting: When α-TCP is mixed with water, it dissolves and the precipitation of nano-sized elongated calcium-deficient hydroxyapatite (CDHA) particles occurs, leading to the hardening of the cement and the increase in mechanical strength [6,7].

Furthermore, ion substitutions in calcium phosphates have become the subject of intense investigations in the last decades [7–11], with the aim of enhancing bioactivity and also gaining the ability to deliver therapeutic ions such as Sr$^{2+}$ directly in situ, which is a promising approach to sustain bone regeneration also in osteoporotic patients, as an alternative to oral administration of drugs. In particular, strontium-doped CPCs [12,13] have shown excellent osteogenic and osteointegrative properties, and a dose-dependent ability to modulate the bone cell fate [14,15].

Nevertheless, CPCs are generally associated with low compressive strength, limited porosity, and poor injectability [5], so new strategies to optimize their processing are

highly desired to make them desirable for clinical applications [16–19]. The performance of CPCs strongly depends on the fine adjustment of multiple parameters, including powder particle size distribution and shape, ion doping, liquid-to-powder ratio (LP), and polymeric additives [17,19,20]. Despite some attempts of modeling the performance of CPCs being reported [19,21,22], even by means of artificial intelligence methods [23], a comprehensive overview on the multifactorial interaction among the CPCs processing parameters is still lacking [19]. The precise investigation about the rheology of CPCs also represents a critical point, due to the inherent heterogeneity of the material. In this respect, some works are focused on the study of the injectability of calcium phosphate pastes based on β-TCP-water systems, mainly because of longer setting times if compared with α-TCP formulations [20,24]. These studies aimed to improve the understanding of phenomena impairing the cohesion of CPCs during extrusion, thanks to their slow hardening kinetics. Nevertheless, those results cannot easily translate to the comprehension of the rheological properties and extrusion ability of any self-hardening formulation [19].

New approaches to improve the injectability and cohesion of CPCs have been mainly investigated by increasing the viscosity of the setting solutions [25,26]; however, the increase in the cement cohesion was often related to significant deviations from the desired setting times and mechanical performance [27]. In this respect, it was observed that smaller CaP precursor particles promote faster transformation into apatite, a more packed structure, and higher mechanical strength [19,28–30]; on the other hand, the introduction of foreign ions into the calcium phosphate structure generally leads to the slackening of the setting times [31,32].

Several approaches were explored to improve the hydraulic reactivity of the powders, including both thermal calcination [33] and different milling treatments [28,34–40]. As regards to the milling process, the high-energy planetary ball milling was also reported as an efficient treatment to rapidly reduce the powder particle size [28,34,39]; however, the correlation between milling time and ball milling diameter has not yet been investigated.

It was reported that the regime of powder-liquid mixing possibly affects the injectability of CPCs, but no studies are present in the literature [19]. In this respect, a direct comparison between low-energy hand mixing and high-energy shear, rotational mixing regimes on both viscosity and the extrusion force of pastes, is proposed in this work.

As a general rule, the optimization of a process by varying one single factor at a time proved to be very limited and time consuming, and novel strategies to obtain an overview of the role of the processing parameters on the performance of CPCs are demanded [19].

Despite the effect of several parameters previously investigated, to the best of our knowledge, the main limitation reported in the literature is the adoption of only two-level factorial design schemes [22,41].

In this respect, the aim of this study was a wide-range characterization of several processing parameters on the final performance of Sr-substituted α-TCP-based CPCs enriched with sodium alginate, previously reported to improve injectability, cohesion, compression strength, and final osteointegration [42], according to three-level Design of Experiment (DoE) schemes, in order to shed light both on the interactions between the parameters [43] while also proposing modeling equations.

## 2. Materials and Methods

Cement Preparation. Sr-doped α-tricalcium phosphate (α-TCP) powders with different strontium content (Sr/(Ca + Sr) = 0, 2, 4 mol%, henceforth coded as Sr0, Sr2, Sr4, respectively) were prepared by solid-state reaction of $CaHPO_4$, $CaCO_3$, and $SrCO_3$ (Merck KGaA, Darmstadt, Germany) [44,45] at three different temperatures (1200 °C, 1400 °C, and 1600 °C) and dwell times (20, 60, 180 min) in a Pt crucible, followed by rapid quenching in air. A more detailed preparation route of similar powders, together with the chemico-physical characterizations proving the substitution of calcium with strontium ions, has been previously reported [42]. Then, the powders were milled in ethanol using a planetary mono mill (Pulverisette 6 classic line, Fritsch, Idar-Oberstein, Germany) at 400 rpm [36,37]

into a zirconia jar with three different milling times (15, 50, and 90 min) and zirconia milling ball diameters (1, 2, and 5 mm) (Nikkato corporation, Osaka, Japan). Afterward, aqueous solutions containing three different amounts of $Na_2HPO_4$ (0, 2.5, and 5 wt%) (Merck KGaA, Darmstadt, Germany) and sodium alginate (0, 1, and 2 wt%) (Alginic Acid Sodium Salt from Brown Algae, Merck KGaA, Darmstadt, Germany) were prepared and mixed by hand with a spatula for about 30 s according to three different liquid-to-powder mass ratios (0.50, 0.55, and 0.60). The sodium alginate was chosen to improve the injectability, cohesion, and compression strength of the cements, without affecting the hardening process [42]. In addition, cylindrical molds (diameter = 6 mm, height = 2 mm) were filled with the as-obtained pastes and kept at 37 °C in saturated humidity conditions to obtain consolidated samples to be tested for their compressive strength.

Cement characterization. The phase composition of the precursor powders and the cements was investigated by X-ray diffraction (2θ range = 10–80°, scan step = 0.02°, step time = 0.5 s) (Bruker D8 Advance Diffractometer equipped with LINXEYE detector, Karlsruhe, Germany); the domain size of the α-TCP crystals was also investigated by means of Rietveld analysis by using the crystal models of α-TCP [46], β-TCP [47], and HA [48] phases. The particle size of the powders (D50) was analyzed with Sedigraph 5100 (Micromeritics, Norcross, GA, USA). The morphology of cements was investigated by scanning electron microscopy (FEG-SEM, Sigma NTS GmbH, Carl Zeiss, Oberkochen, Germany). The cement injectability was qualitatively evaluated by measuring the amount of cement in the syringe after extrusion [29]. The initial and final setting times of the cements were evaluated with Gillmore apparatus, according to standard ASTM C266-99, by testing 5 specimens for each formulation. The compressive strength and Young's modulus of the cements were evaluated by testing 5 cylindrical specimens (diameter = 8 mm, height = 17 mm) after 3 days of immersion in Simulated Body Fluid (SBF) [49] at 37 °C, followed by drying for 24 h in a vacuum chamber. The tests were performed in agreement with ISO 9917 with a universal testing machine (MTS Insight 5, Eden Prairie, MN, USA). Peirce's criterion was applied to exclude outliers [50]. Young's modulus was calculated by evaluating the slope of the linear regression of the stress–strain curves until the failure stress of the specimen. The porosity was evaluated on the cylindrical specimens as $P = 1 - \rho/\rho 0$, where ρ is the cement density determined as a weight-to-volume ratio and ρ0 refers to the actual phase composition, as obtained by the Rietveld analysis. The viscosity measurements were performed by placing about 2 mL of cement into a rotational rheometer (Bohlin C-VOR 120, Malvern Instruments, United Kingdom) after 45 s of mixing; the setup included parallel plates with a 20 mm upper plate diameter, a gap distance of 750 μm, and application of a constant shear rate of 50 s$^{-1}$ for 2 min, as reported during 3D-printing extrusion [51]. The effect of mixing energy on both extrusion force and viscosity was evaluated by mixing Sr2 powders planetary milled for 50 min with 2 mm diameter balls with 5 wt% sodium phosphate solutions, enriched with 2 wt% sodium alginate at a constant LP value of 0.60. The pastes were mixed both manually, using a bowl and spatula arrangement, or automatically, using a high-energy planetary shear-mixer (Thinky Mixer ARE-500, Thinky, Japan) at 1000 rpm. Two different mixing times were explored, 45 and 90 s. After mixing, the pastes were carefully loaded into 20 mL Luer lock syringes (Troge Medical, Hamburg, Germany) with an 18.8 ± 0.1 mm inner barrel diameter (DBARREL) and length of nozzle LNOZZLE = 9.4 ± 0.1 mm. In addition, a needle was used (needle length 20 mm, needle diameter 0.3 mm). Using a Zwick/Roell Z050 Universal Testing Machine, pastes were extruded at a constant plunger rate (40 mm/min) until the maximum force of 400 N was reached. The extrusion force profiles basically include two phases: an initial phase derived from the plunger friction and the material compacting before paste extrusion and a second phase derived from the force needed to extrude the material. For each condition, three specimens were tested and only the s extrusion force of the second phase values were considered.

Statistical methods. Three different face-centered composite schemes (called Powder Thermal Treatment, Powder Milling, and Setting Solution) were implemented (Table 1).

For each scheme, the selected input and output parameters are summarized in the Factor and Output columns, respectively, together with the coded levels and symbols.

**Table 1.** Input factors and correspondent coded levels.

| | Factor | Factor Symbol | Coded Levels | | | Output | Output Symbol |
|---|---|---|---|---|---|---|---|
| | | | −1 | 0 | +1 | | |
| Powder Thermal Treatment | Sr/(Ca + Sr) (mol%) | Sr | 0 | 2 | 4 | • $\alpha$-TCP (wt%) <br> • $\alpha$-TCP domain size (nm) <br> • Initial setting time (min) * <br> • CDHA after 24 h at 37 °C (wt%) * | $\alpha$ <br> $\alpha$Ds <br> $t_{in}$ <br> HA |
| | Temperature (°C) | T | 1200 | 1400 | 1600 | | |
| | Dwell time (min) | Dt | 20 | 60 | 180 | | |
| Powder Milling | Sr/(Ca + Sr) (mol%) | Sr | 0 | 2 | 4 | • Average particle size (µm) <br> • Initial setting time (min) ** <br> • Final setting time (min) ** | $D_{50}$ <br> $t_{in}$ <br> $t_{fin}$ |
| | Milling Time (min) | MT | 10 | 50 | 90 | | |
| | Milling Balls Diameter (mm) | MBD | 1 | 2 | 5 | | |
| Setting Solution | Na$_2$HPO$_4$ (wt%) | SP | 0 | 2.5 | 5 | • Initial setting time (min) $^\Delta$ <br> • CDHA after 24 h at 37 °C (wt%) $^\Delta$ <br> • Viscosity (Pa·s) $^\Delta$ | $t_{in}$ <br> HA <br> $\eta$ |
| | Sodium Alginate (wt%) | SA | 0 | 1 | 2 | | |
| | Liquid-to-Powder ratio (-) | LP | 0.50 | 0.55 | 0.60 | | |

*: Cements prepared with powders milled for 15 min with 5 mm diameter balls, and solutions containing 5 wt% Na$_2$HPO$_4$, no sodium alginate (L/P = 0.55); **: Cements prepared with powders synthesized at 1400 °C for 60 min, and solutions containing 5 wt% Na$_2$HPO$_4$, no sodium alginate (L/P = 0.55); $^\Delta$: Cements prepared with Sr2 powders synthesized at 1400 °C for 60 min, milled for 50 min with 2 mm diameter balls.

The number of experiments required for each scheme, n, is given by the formula n = $2^k$ + 2k + c, where k is the number of factors and c is the number of center points [52,53]. Three center points were adopted for each scheme: The resulting 17 experiments were performed in triplicate in random order.

The regression analysis of data was performed according to the second-order polynomial: $y = \beta_0 + \sum \beta_i x_i + \sum_{i<j} \sum \beta_{ij} x_i x_j + \sum \beta_{ii} x_i^2$, where $y$ is the response; $x_i$ and $x_j$ are the coded factor levels; and $\beta_0$, $\beta_i$, $\beta_{ii}$, and $\beta_{ij}$ are the mean values of constant, linear, quadratic, and interaction coefficients, respectively. As a general rule, to maximize a response, the factors with positive coefficients should be increased and those with negative coefficients should be decreased. Bidimensional contour plots were generated with Matlab software, providing a graphical overview about the interaction between the parameters: The output variables are represented from blue (lower values) to red (higher values) colors, as reported in the colored scale bars. Moreover, the more the contour lines diverge from horizontal or vertical trends, the more significant the interaction. The experimental data were statistically evaluated by two-way analysis of variance (ANOVA), followed by Tukey's multiple comparisons test (GraphPad Prism 6 software).

## 3. Results

### 3.1. Powder Thermal Treatment

According to the face-centered composite design, the results of the Powder Thermal Treatment scheme are reported in Tables S1 and S2 and Figure 1.

Considering only the significant coefficients (*p*-value, *p* < 0.05), the following regression equations can be obtained:

$$\alpha = 87.53 + 24.63 \, T - 16.11 \, T^2 - 5.57 \, Sr + 4.91 \, Sr \cdot T \tag{1}$$

$$\alpha Ds = 440.54 - 89.21 \, T^2 - 87.20 \, Sr + 84.00 \, T - 78.75 \, Sr \cdot T \tag{2}$$

$$t_{in} = 26.22 + 14.40 \, Sr - 13.20 \, T \tag{3}$$

$$HA = 29.35 + 13.10 \, T - 11.70 \, Sr \tag{4}$$

As reported in Equation (1), higher temperatures of the solid-state reaction (T) favored the formation of $\alpha$-TCP, while the incorporation of strontium into the TCP lattice induced the stabilization of $\beta$-TCP polymorph and the formation of smaller $\alpha$-TCP crystal domains, as previously reported [42,54], especially at lower temperatures (Figure 2a).

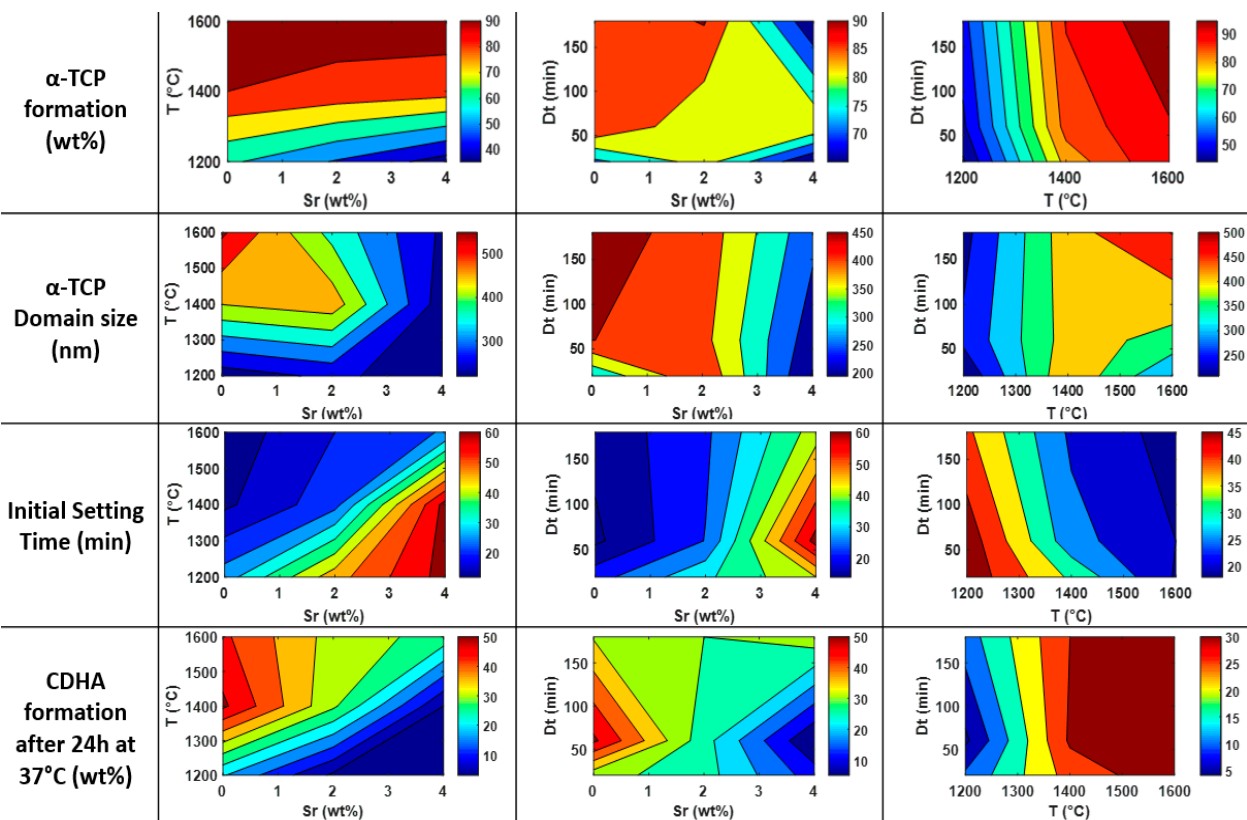

**Figure 1.** Contour plots related to the Powder Thermal Treatment scheme: The effect of strontium amount (Sr), temperature (T), and dwell time (Dt) on the α-tricalcium phosphate (α-TCP) formation (first row), α-TCP domain size (second row), initial setting time (third row), and calcium-deficient hydroxyapatite (CDHA) formation (forth row). The output variables are represented, as reported in the colored scale bars, from blue (lower values) to red (higher values).

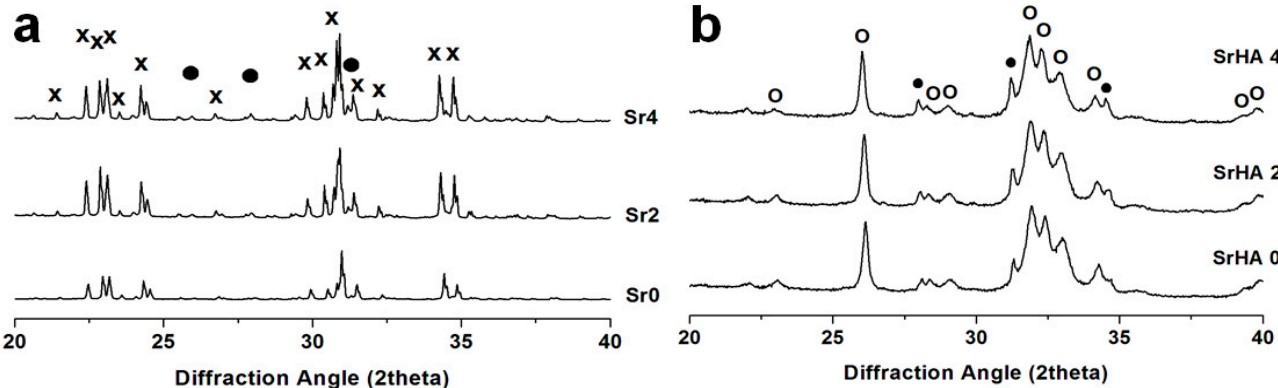

**Figure 2.** XRD patterns of Sr0, Sr2, and Sr4 powders prepared (**a**) and the correspondent Sr-containing apatite (SrHA) cements after 7 days of immersion in SBF at 37 °C (**b**). Peaks legend: α-TCP (X), β-TCP (●), CDHA (o).

Upon mixing with the solution, the strontium substitution was the main factor delaying the setting times of the cements, while a quicker transformation of α-TCP into CDHA was observed after 24 h for the powders obtained at higher temperatures. After 7 days, the complete transformation of α-TCP into CDHA occurred, with no substantial changes in the amount of β-TCP among the formulations (Figure 2b).

The microstructure of cements with different amounts of strontium (Sr0 and Sr4) and high-temperature of synthesis (1200 °C and 1600 °C) was investigated and compared at 24 h after setting and immersion at 37 °C, considering two different dwell times of high-temperature powder synthesis before quenching, 15 (Figure 3) and 180 min (Figure 4).

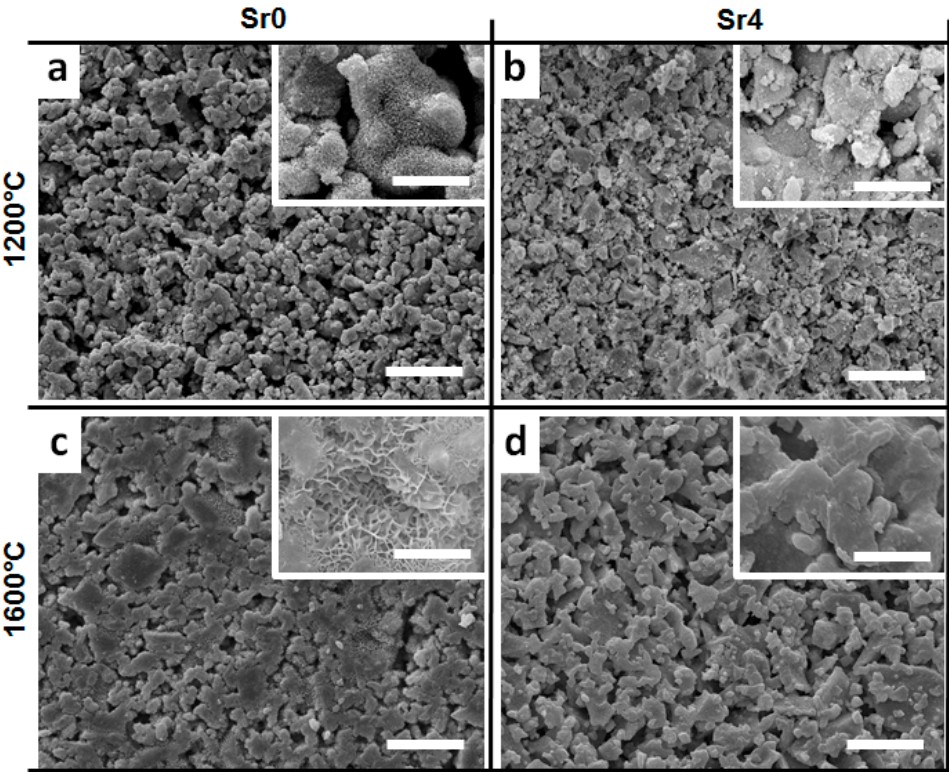

**Figure 3.** Microstructures of different cement formulations with dwell time of high-temperature powder synthesis equal to 15 min: (**a**) Sr0, 1200 °C; (**b**) Sr4, 1200 °C; (**c**) Sr0, 1600 °C; (**d**) Sr4, 1600 °C after 24 h at 37 °C. Scale bar: Image = 20 μm, insets = 5 μm.

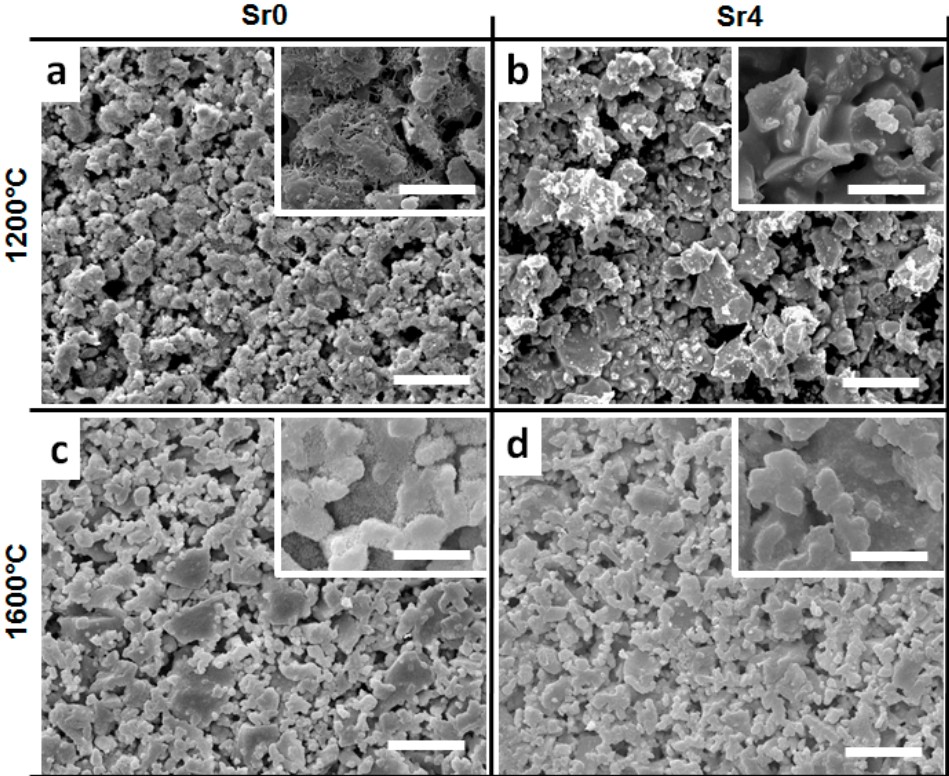

**Figure 4.** Microstructures of different cement formulations with dwell time of high-temperature powder synthesis equal to 180 min: (**a**) Sr0, 1200 °C; (**b**) Sr4, 1200 °C; (**c**) Sr0, 1600 °C; (**d**) Sr4, 1600 °C after 24 h at 37 °C. Scale bar: Image = 20 μm, insets = 5 μm.

A larger amount of needle- to flaky-like apatitic structures were observed for the strontium-free cements, as well as the cements prepared with powders synthesized at higher temperature, especially in the case of limited dwell time of high-temperature powder synthesis (Figure 3a,c).

On the basis of adequate phase composition and hydraulic reactivity (see Figure 1 and Table S1), the Sr2 formulation treated at 1400 °C for 1 h was selected for further investigations.

### 3.2. Powder Planetary Milling

The results of the Powder Planetary Milling scheme are reported in Tables S3 and S4 and Figure 5.

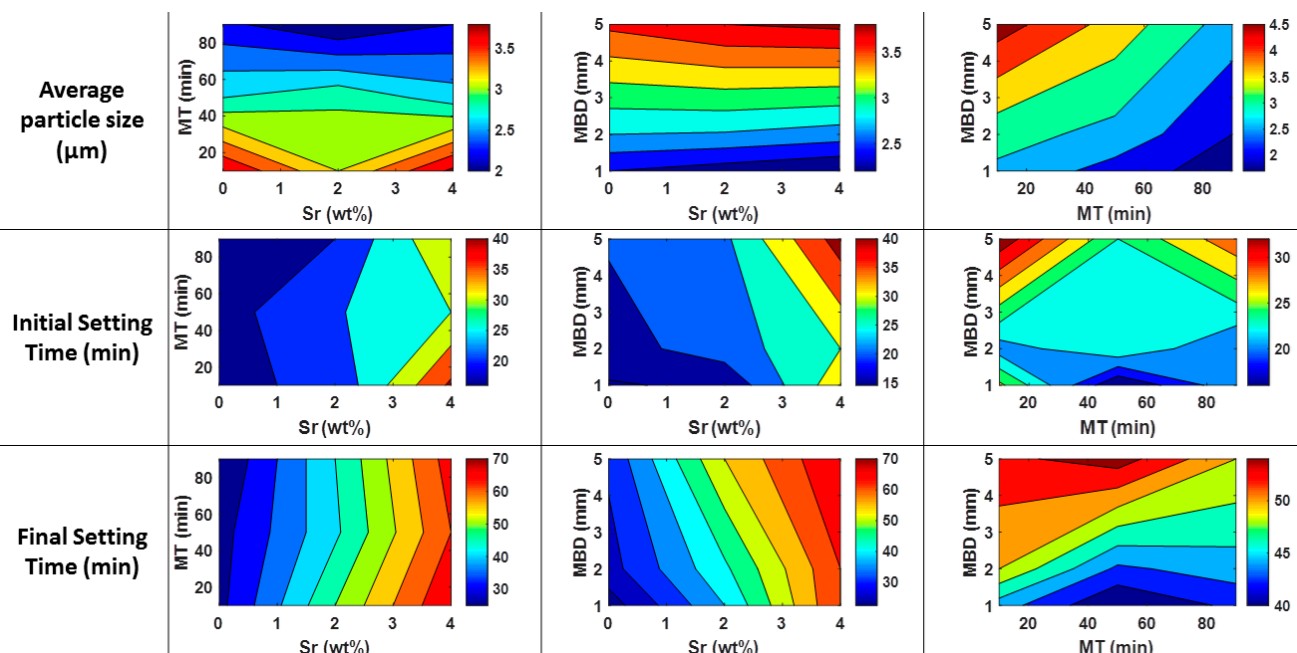

**Figure 5.** Contour plots related to the Powder Milling scheme: The effects of strontium amount (Sr), milling time (Mt), and milling balls diameter (MBD) on the average particle size (first row), and initial and final setting times (second and third row, respectively). The output variables are represented, as reported in the colored scale bars, from blue (lower values) to red (higher values).

Considering only the significant coefficients ($p < 0.050$), the following regression equations can be obtained:

$$D50 = 2.80 - 0.76\,MT + 0.73\,MBD + 0.32\,MBD^2 - 0.20\,MT{\cdot}MBD \tag{5}$$

$$t_{in} = 21.86 + 9.10\,Sr + 4.30\,MBD \tag{6}$$

$$t_{fin} = 42.87 + 19.30\,Sr + 4.40\,MBD \tag{7}$$

It was observed that both prolonged milling times and smaller milling balls mainly reduced the average particle size of the powders, also with significant interactions between the factors. Conversely, the setting times of the cements were basically slackened by increasing the strontium doping or the milling ball diameter.

In addition to the reduction in average particle size, smaller milling balls also resulted in the increase in the full-width at half-maximum (FWHM) of the peaks, indicating the reduced crystal size of the powder (Figure S1), according to the Scherrer equation [39].

On this basis, we also evaluated the compressive strength of cements by varying both the amount of strontium and the milling balls diameter: The compressive strength

significantly increased with decreasing milling balls diameter (MBD) ($p < 0.001$), while the higher values were exhibited by the Sr2 formulation ($p < 0.05$) (Figure 6).

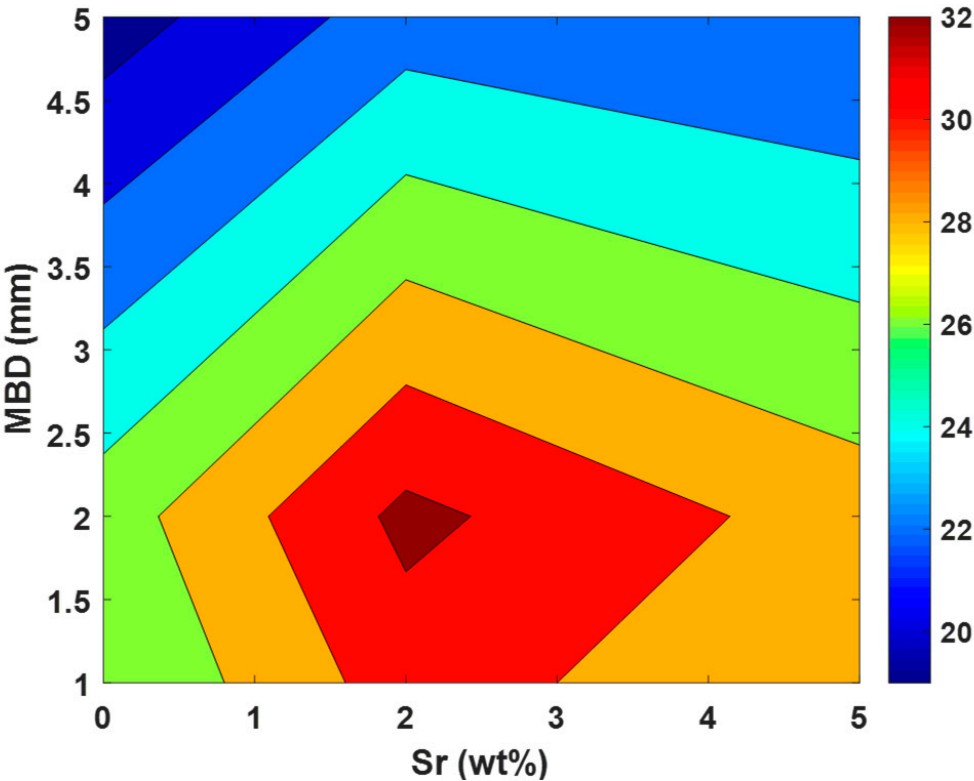

**Figure 6.** Contour plot related to the effect of strontium amount (Sr) and milling balls diameter (MBD) on the compressive strength of the cements, reported in the colored scale bar, from blue (lower values) to red (higher values).

A flaky- to needle-like morphology was detected for all the formulations, with a prevalence of more homogeneous sharp needle-structures when using smaller milling balls (Figure 7). The porosity range was 45–50%.

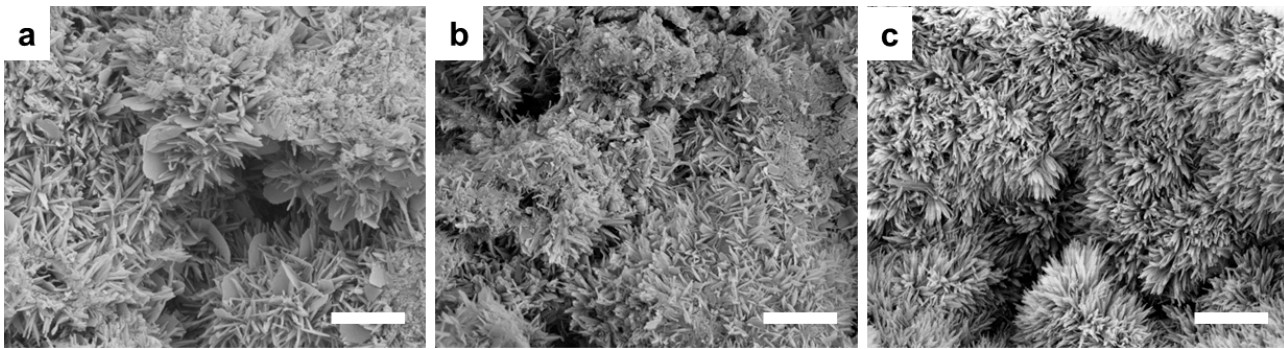

**Figure 7.** Morphology of the Sr2 cements obtained by powders milled with different milling media diameters: 5 (**a**), 2 (**b**), 1 mm (**c**). Scale bars: 1 μm.

The Sr2 powder milled for 50 min with 2 mm diameter balls was selected for further investigation, due to the adequate hydraulic reactivity and compression strength.

### 3.3. Setting Solution and Mixing

The results of the Setting Solution and Mixing scheme are reported in Tables S5 and S6 and Figure 8.

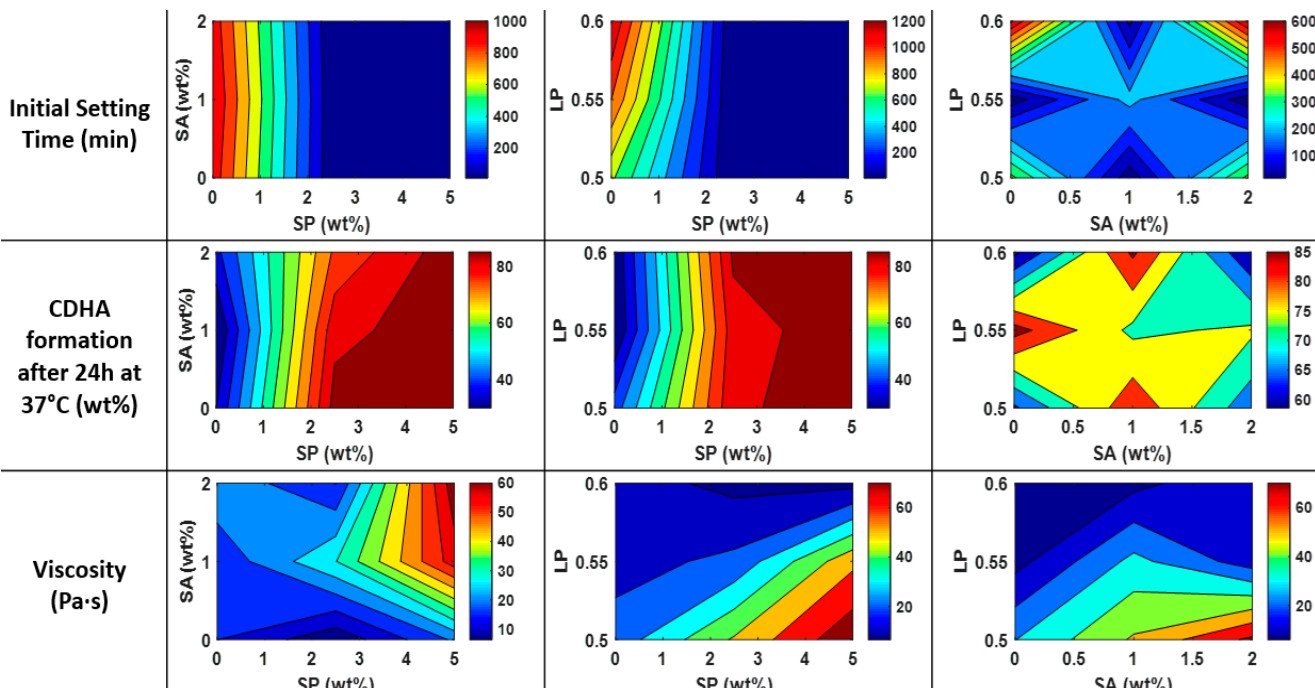

**Figure 8.** Contour plots related to the Setting Solution and Mixing scheme: The effects of sodium phosphate (SP), sodium alginate (SA), and liquid-to-powder ratio (LP) on the initial setting time (first row), powder hydraulic reactivity (second row), and viscosity (third row). The output variables are represented, as reported in the colored scale bars, from blue (lower values) to red (higher values).

Considering specifically the coefficients with $p < 0.05$, the following regression equations can be obtained:

$$t_{in} = 30.62 - 477.70\,SP + 473.41\,SP^2 - 118.50\,SP \cdot LP + 99.10\,LP \qquad (8)$$

$$HA = 82.74 + 26.20\,SP - 23.05\,SP^2 \qquad (9)$$

$$\eta = 26.96 - 20.72\,LP - 15.52\,SP\,LP + 13.15\,SP + 11.08\,SA \qquad (10)$$

As observed in Equation (8), the initial setting times of the cements were mainly reduced with the increase in the sodium phosphate amount (SP), even more than reducing the liquid-to-powder ratio (LP). Conversely, the transformation into CDHA after 24 h at 37 °C was basically promoted by increasing SP.

The reduction in LP mainly increased the viscosity, but also, SP and sodium alginate (SA) played significant roles; in this respect, it should be noted that, with increasing LP, the viscosity steadily waned even in correspondence of the highest amount of SP and SA.

### 3.4. Effect of Different Mixing Methods

The extrusion force profiles are compared in Figure 9. The profile obtained with an empty syringe (ES sample) was determined and presented a steady decrease in extrusion pressure nearing the end of the test [24]. The mean extrusion force of Phase 2 was determined for each sample; as shown in the inset, both mixing times and procedures led to extrusion forces that can be applied by the surgeon to allow injection through a cannulated needle (reported as 100 [55] to 300 N [56]). Nevertheless, a significant reduction in the extrusion force was detected while increasing the mixing time (from 45 to 90 s), and mostly changing from hand-spatula mixing (sample S) to the automatic planetary mixer (samples M). The M samples exhibited both moderately dispersed force profiles of Phase 2, as well as complete injectability, as also confirmed by the quasi-overlapping profiles with the ES sample curve after the completion of Phase 2.

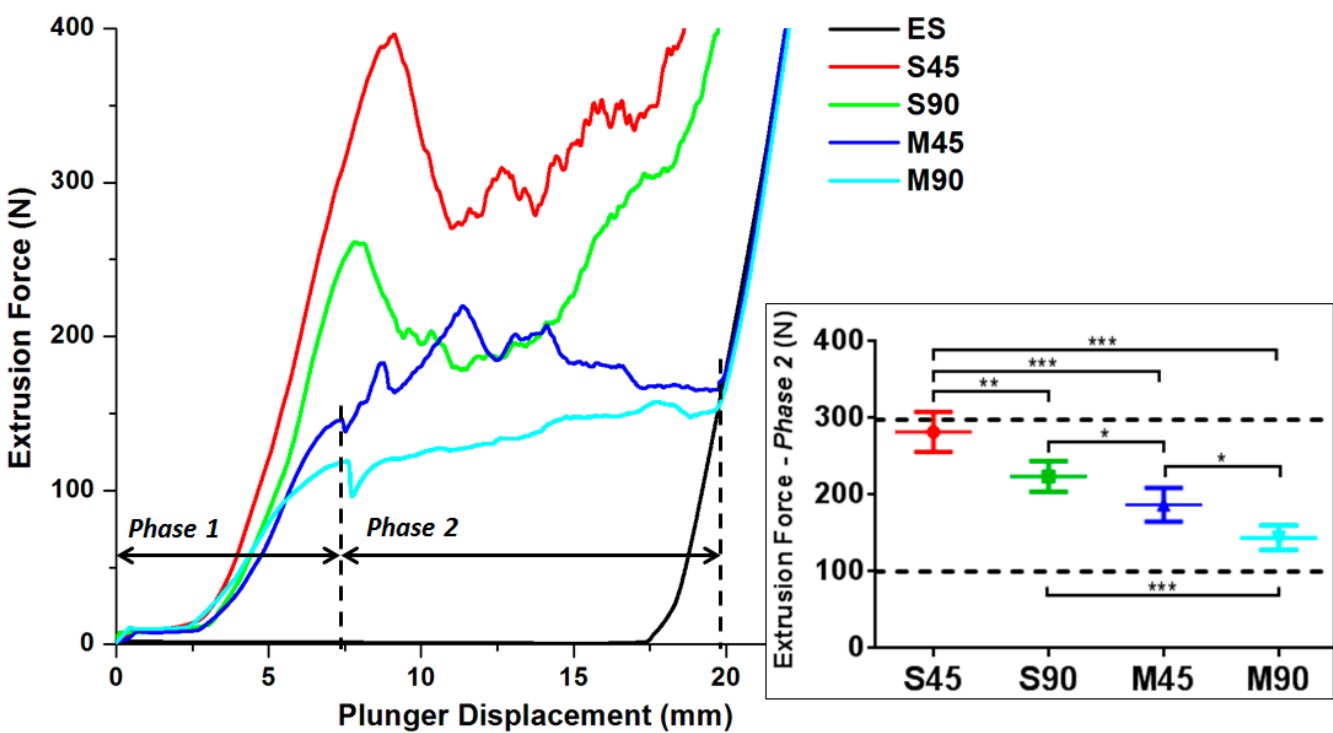

**Figure 9.** Effect of different mixing times and shear regime on extrusion force–plunger displacement profiles (ES: Empty syringe, S: Spatulated samples, M: Planetary-mixed samples, "45" or "90" refers to the correspondent mixing time, in seconds). Phase 1 profile includes several contributions, such as the plunger friction and the material compacting before paste extrusion. Phase 2 indicates the force profile during the extrusion of the material. Inset: Extrusion force calculated for each test, with the reported range for manual orthopedic surgeon injection in dotted line (*: $p < 0.05$, **: $p < 0.01$, ***: $p < 0.001$).

In this test, both mixing times and procedures did not significantly affect the setting times of the formulations, in the range tin = $15 \pm 3$ min and tfin = $29 \pm 4$ min.

Given the complex interaction of the processing parameters, Table 2 provides a qualitative summary of the findings of the present study.

**Table 2.** The qualitative correlation of each increasing factor on the outputs is reported as positive (arrows up), negative (arrow down), statistical significance (double arrow), or negligible statistical significance (~).

| | Outputs | | | | | | |
|---|---|---|---|---|---|---|---|
| **Increasing Input Factor** | **α-TCP Amount** | **α-TCP Domain Size** | **Average Particle Size** | **Initial Setting Time** | **Final Setting Time** | **HA Amount** | **Viscosity** |
| Temperature | ↑↑ | ↑↑ | | ↓ | | ↑↑ | |
| Dwell Time | ~ | ~ | | ~ | | ~ | |
| Strontium Amount | ↓ | ↓↓ | ~ | ↑↑ | ↑↑ | ↓ | |
| Milling Time | | | ↓↓ | ~ | ~ | | |
| Milling Balls Diameter | | | ↑ | ↑ | ↑ | | |
| Sodium Phosphate | | | | ↓↓ | | ↑↑ | ↑ |
| Sodium Alginate | | | | ~ | | ~ | ~ |
| Liquid-to-powder ratio | | | | ↑ | | ~ | ↓ |

## 4. Discussion

The present study provides an overview about the role of a multitude of processing parameters on the rheological and mechanical performance of strontium-doped CPCs, according to multiple three-level Design of Experiment (DoE) schemes. In this way, the simultaneous variation of three factors was analyzed, in order to evaluate both the linear and the quadratic interactions between the factors [43] while also proposing modeling equations.

The previous literature reported contradicting results about the performance of CPCs [13,19], probably ascribable to the difficulty of investigating the role of processing parameters by varying only one factor at a time.

The main parameter relevant for the manipulation and characterization of CPCs is the setting time, which depends on a complex interaction between many factors.

For example, several studies in the literature reported the preparation of self-hardening CPCs by hydrolysis of $\alpha$-TCP precursors, but only reporting a specific thermal treatment for the synthesis of $\alpha$-TCP [28,33,37,57–59]. In this respect, it was observed that the firing step greatly affects the final reactivity of the $\alpha$-TCP upon mixing with liquid solutions [13]. The present work demonstrates that, rather than the dwell time, the temperature selected for the synthesis of $\alpha$-TCP is the main factor determining the rate of transformation of the cement into CDHA. However, the attainment of pure Sr-doped $\alpha$-TCP is hampered by the presence of strontium (Figure 2), as it promotes the formation of $\beta$-TCP polymorph [60]. In this respect, we found that the partial substitution of $Ca^{2+}$ with $Sr^{2+}$ ions is the predominant factor affecting the $\alpha$-TCP crystal size and inducing both a significant decrease of slackening of the setting times [13,32,42].

This study also showed that higher hydraulic reactivity of the precursor powders (i.e., higher $\alpha$-TCP amount, in respect of $\beta$-TCP) favors the achievement of nanostructured morphology, represented by sharp needle-like apatitic crystals developed upon setting at 37 °C (Figures 3 and 4). In particular, higher sintering temperatures led to an increase in crystal growth ($\alpha$Ds), resulting in a powder with reduced specific surface area and slower rate of dissolution and precipitation to CDHA, while the introduction of strontium induced a retarded hydration of $\alpha$-TCP as observed by calorimetry studies [54]. The kinetics of the hydration reaction of the precursor powder basically affects the morphology of the final apatitic crystals: The faster the dissolution of the $\alpha$-TCP powder, the faster the supersaturation of the solution with respect to calcium and phosphate ions, causing the formation of small and highly entangled needle- to flaky-like apatitic crystals, whereas in the case of strontium-doped formulation, the delayed setting produced larger, poorly entangled crystals. In this respect, the kinetics of the hydration process affects the nanostructure of the CPCs, thus tailoring both setting times and mechanical performance. On this basis, the thermal treatment at 1400 °C for 1 h was chosen to maximize the amount of $\alpha$-TCP (i.e., the hydraulic reactivity of the powders), while preventing the microstructural coarsening reported in the case of longer firing steps [59].

The hydrolysis of the $\alpha$-TCP powders is also influenced by the particle size distribution [61]. In this work, the average particle size and the crystallinity of the powders was mainly reduced by increasing the milling time and decreasing the milling media diameter (Figure 5 and Figure S5), possibly due to the higher surface area exposed by the milling spheres [62,63]. Conversely, both the initial and setting times were basically slackened by the strontium doping, if compared with milling parameters. Interestingly, powders milled with smaller media exhibited significantly reduced hardening times and have given rise to cements with improved compressive strength (Figure 6). In this work, the Sr2 formulations obtained by milling with 2 mm diameter balls exhibited a prevalence of sharper needle-like crystals (Figure 7) and the highest compressive strength, even comparable to that of previously reported apatite-cement formulations reinforced with fibers [17]. It was hypothesized that the increase in the strontium content induced an increase in the crystal defects, leading to the slackening of hydrolysis and setting processes and to reduced strength [64].

A critical aspect in the setting behavior of cements is also given by rheological properties. The major role of the setting solution composition and liquid-to-powder ratio on the rheology of CPCs was quantified in Equation (10), while only being qualitatively reported in previous studies [13,17,19]. In particular, the addition of sodium alginate improved the flowability and injectability of the cement without interfering with the hydrolysis of the powder and the setting times, as previously observed [42]. In this respect, the addition of a small amount of alginate enhanced the setting times and the production of a much denser microstructure, as well as higher mechanical performance, possibly ascribable to the

cross-linking of sodium alginate by $Ca^{2+}$ released from α-TCP [65], while higher amounts of alginate were even reported as setting delayers [66]. The formation of precipitated HA was mainly favored by increasing the sodium phosphate concentration (Figure 8). Deep rheological investigation of the early stages of the reaction yielding the CPCs hardening has been limited so far probably due to the difficulties in measuring a very large viscosity range and to the problems of measuring the real material viscosity rather than the interactions between the material and the rheometer during the test [36]. For that reason, only qualitative viscosity ranges are reported (e.g., 100–1000 Pa·s for 10 min) [20,67]. In this work, the viscosity of some cement formulations has been determined by applying the constant shear rate typically reported during 3D-printing extrusion [51]. The reduction in the liquid-to-powder ratio was the main factor increasing the viscosity of the cements (Figure 8).

What was claimed was the lack of studies comparing the effect of different mixing energies on the rheology of CPCs [19]; the only research cited in this respect involved different mixing shear regimes on the rheology of construction cements [68]. They found improved flow properties of cements subjected to high shear mixing, compared to the same cement mixed with a low-shear spatula. With the purpose of shedding light on this aspect related to CPCs, this work compares for the first time low-shear (hand-mixing) with planetary high-shear mixing on the extrusion force requested for delivering the material from cannulated syringes. We found that a significant decrease in the mean extrusion force of the steady phase (see Phase 2 in Figure 9) occurring by both prolonging the mixing time and using high-shear planetary mixing. Moreover, a minor dispersion in the data on the extrusion force was observed for the planetary mixed samples, if compared to the manually mixed ones, but no significant differences among the formulations were detected, in terms of setting rates. This result suggests that by increasing the mixing energy, a more efficient interaction of the liquid with the powder occurred, preventing uncontrolled powder agglomeration and improving the CPC flowability, without altering the setting reaction kinetics. This finding becomes particularly interesting toward the improvement in material extrusion from cannulated syringes, relevant for clinical procedures, as well as for 3D printing technology, where the layer-by-layer deposition of CPCs plays an important role [69].

In this work, the regression equations exhibited a good predictability of the outputs, as evidenced by the high coefficients of determination, R2 and adjusted R2 (R2-adj). In addition, the nonlinear terms (e.g., quadratic or interaction) confirm the complexity and heterogeneity of the CPCs processing to obtain specific performance, but also provide a quantitative overview about the effect of each processing parameter, if compared with the qualitative results previously reported in the literature. This work examines the three-level design of experiments as a useful tool toward the preparation of injectable and self-hardening CPCs addressed to specific clinical application.

## 5. Conclusions

The present work provides an overview about the role of thermal treatment of powders, and milling and mixing processes on the final performance of Sr-doped CPCs, for the first time by three-level Design of Experiments schemes. The results quantitatively demonstrated that the hydraulic reactivity of the precursor phase is mainly affected by the temperature, while the strontium doping basically slackened the cement hardening. Besides, a significant reduction in setting times and increase in compressive strength by decreasing the powder milling balls' diameter were observed, due to the resulting nanostructure of the material. The cement viscosity was mainly affected by variations in the liquid-to-powder ratio. Interestingly, a significant decrease in extrusion force occurred by high-energy mixing and longer mixing time, also paving new strategies to improve the extrusion of calcium phosphate injectable pastes. As quantitative modeling equations are also proposed, the present work includes a useful approach to shed light on the compli-

cated processing of CPCs, toward the design of injectable formulations with customable features, depending on the clinical need.

**Supplementary Materials:** The following are available online at https://www.mdpi.com/2076-341 7/11/5/2075/s1, Figure S1: XRD patterns of Sr2 powders prepared with different MBD (5 mm, 2 mm, 1 mm), with respective FWHM of the highest peak (i.e., $2\vartheta$ =30.71), Table S1: Results of the Powder Thermal Treatment scheme, Table S2: Results of the regression analysis of the Powder Thermal Treatment scheme, Table S3: Results of the Powder Thermal Treatment scheme, Table S4: Results of the regression analysis of the Powder Planetary Milling scheme, Table S5: Results of the Setting solution and mixing scheme, Table S6: Results of the regression analysis of the Setting solution and mixing scheme.

**Author Contributions:** Conceptualization, M.D. and S.S.; methodology, M.D.; investigation, M.D.; resources, A.T.; data curation, M.D.; writing—original draft preparation, M.D.; writing—review and editing, M.D., D.G. and S.S.; supervision, A.T. and S.S. All authors have read and agreed to the published version of the manuscript.

**Funding:** The research leading to these results has received funding from the EU project NMP3-SL-2010-SMALL-3-246373 (OPHIS) and the National Project, PNR-CNR Aging Program 2012–2014.

**Acknowledgments:** The authors wish to acknowledge Cesare Melandri for the execution of mechanical and extrusion force tests.

**Conflicts of Interest:** The authors declare that there is no conflict of interest regarding the publication of this article.

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
