# Peer review of "Nanostructured Strontium-Doped Calcium Phosphate Cements: A Multifactorial Design"

_applsci, doi:10.3390/app11052075_

Round 1

Reviewer 1 Report

In my humble opinion, this manuscript might me accepted after a minor revision:

  1. Both subscripts and superscripts are not used for chemical formulae.

  1. It is not clear how the colored contour plots were created from the results of 17 experiments performed in triplicate. Please, clarify.

  1. Refs. 35 and 42: volume and pages are missing.

Author Response

Point 1. Both subscripts and superscripts are not used for chemical formulae.

Response 1. Thanks for the suggestion, we modified the chemical formulae accordingly.

Point 2. It is not clear how the colored contour plots were created from the results of 17 experiments performed in triplicate. Please, clarify.

Response 2. Each experiment (e.g. each row of the Supplementary Tables) was performed in triplicate, while the reported 2D contour plot were actually created by averaging the output values corresponding to each combination of inputs. In particular, the ”contourf” function of Matlab software was then used to obtain the plots.

Point 3. Refs. 35 and 42: volume and pages are missing.

Response 3. Thanks for the comment, we improved the Reference section, accordingly.

Reviewer 2 Report

The manuscript "Nanostructured strontium-doped calcium phosphates cements: a multifactorial design" describes the influence of several important processing parameters on the preparation of strotium-doped calcium phosphate cements, using for the first time three-levels factorial design. Considering the importance of calcium phosphates for developing novel, multifunczional bone regeneration materials, this paper will be of high interest for a broad public. In addition, the paper is well conceived and clearly written.

However, there is a major remark that needs to be addressed befor the publishing. No PXRD results for the cements obtained by powder milling, as well as PXRD and SEM results for the cements obtained by setting solution procedure are presented. I would advise to includ them, at least in supplementary material.

Minor remarks:

  • line 68 - please check the formula for the number of experiments
  • caption of figure 2 - please specify in which media the samples were immersed

Author Response

Point 1. However, there is a major remark that needs to be addressed befor the publishing. No PXRD results for the cements obtained by powder milling, as well as PXRD and SEM results for the cements obtained by setting solution procedure are presented. I would advise to includ them, at least in supplementary material.

Response 1 - Thanks for the comment and suggestion. A new XRD figure was implemented as “Suppl.Figure 5”, showing the effect of milling ball diameter (MBD) on the FWHM of the highest peak of Sr2 formulation. The broadening of the peaks was also commented in the Results section. SEM images describing the morphology of cements prepared with different MBD are actually reported in Figure 7.

Point 2. line 68 - please check the formula for the number of experiments

Response 2. Thanks for the suggestion, we checked and corrected the formula: “…the formula n = 2k + 2k + c, where k is the number of factors …”

Point 3. caption of figure 2 - please specify in which media the samples were immersed

Response 3 - The samples were immersed for 7 days of immersion in SBF at 37°C. We modified the caption, accordingly.

Round 2

Reviewer 2 Report

The authors have addresses all the raised questions adequately.